# Feasibility and Acute Toxicity of Hypo-Fractionated Radiotherapy on 0.35T MR-LINAC: The First Prospective Study in Spain

**DOI:** 10.3390/cancers16091685

**Published:** 2024-04-26

**Authors:** Daniela Gonsalves, Abrahams Ocanto, Eduardo Meilan, Alberto Gomez, Jesus Dominguez, Lisselott Torres, Castalia Fernández Pascual, Macarena Teja, Miguel Montijano Linde, Marcos Guijarro, Daniel Rivas, Jose Begara, Jose Antonio González, Jon Andreescu, Esther Holgado, Diego Alcaraz, Escarlata López, Maia Dzhugashvli, Fernando Lopez-Campos, Filippo Alongi, Felipe Couñago

**Affiliations:** 1Department of Radiation Oncology, Hospital Universitario San Francisco de Asís, GenesisCare, 28002 Madrid, Spain; abrahams.ocanto@genesiscare.es (A.O.); lisselott.torres@genesiscare.es (L.T.); castalia.fernandez@genesiscare.es (C.F.P.); macarena.teja@genesiscare.es (M.T.); miguel.montijano@genesiscare.es (M.M.L.); marcos.guijarro@genesiscare.es (M.G.); felipe.counago@genesiscare.es (F.C.); 2Department of Radiation Oncology, Hospital Universitario Vithas La Milagrosa, GenesisCare, 28010 Madrid, Spain; eduardo.meilan@genesiscare.es (E.M.); alberto.gomez@genesiscare.es (A.G.); maia.dzhugashvli@genesiscare.es (M.D.); fernando.lopez@genesiscare.es (F.L.-C.); 3Facultad de Medicina Salud y Deporte, Universidad Europea de Madrid, 28670 Madrid, Spain; 4Department of Radiation Oncology, GenesisCare Málaga, 29018 Madrid, Spain; daniel.rivas@genesiscare.es (D.R.); jose.begara@genesiscare.es (J.B.); escarlata.lopez@genesiscare.es (E.L.); 5Department of Radiation Oncology, GenesisCare Sevilla, 41092 Madrid, Spain; jose.gonzalez@genesiscare.es; 6Department of Radiation Oncology, GenesisCare Cordoba, 14012 Madrid, Spain; jon.andreescu@genesiscare.es; 7Department of Medical Oncology, Hospital Universitario San Francisco de Asís, GenesisCare, 28002 Madrid, Spain; esther.holgado@genesiscare.es (E.H.); diego.alcaraz@genesiscare.es (D.A.); 8Advanced Radiation Oncology Department, Cancer Care Center, IRCCS Sacro Cuore Don Calabria Hospital, 37024 Verona, Italy; filippo.alongi@sacrocuore.it; 9Radiation Oncology School, University of Brescia, 25121 Brescia, Italy

**Keywords:** adaptive radiotherapy, stereotactic body radiotherapy, workflow

## Abstract

**Simple Summary:**

This study assessed hypo-fractionated radiotherapy’s feasibility and acute toxicity using the first Spanish 0.35T MR-LINAC in 37 patients. Prostate tumors (59.46%) were the most treated, followed by pancreatic tumors (32.44%). Treatment adaptation was successful, with manageable acute toxicity profiles. For prostate cancer, hypo-fractionated radiotherapy yielded promising outcomes with minimal toxicity, predominantly grade I and II cystitis. Pancreatic cancer patients received ablative dose radiotherapy with acceptable toxicity. Quality assurance measures demonstrated precise dose delivery. Overall, our study highlights the safety and feasibility of hypo-fractionated radiotherapy on a 0.35T MR-LINAC, particularly for challenging anatomical sites like prostate and pancreatic tumors, supporting its potential as an effective cancer treatment strategy.

**Abstract:**

This observational, descriptive, longitudinal, and prospective basket-type study (Registry #5289) prospectively evaluated the feasibility and acute toxicity of hypo-fractionated radiotherapy on the first 0.35T MR-LINAC in Spain. A total of 37 patients were included between August and December 2023, primarily with prostate tumors (59.46%), followed by pancreatic tumors (32.44%). Treatment regimens typically involved extreme hypo-fractionated radiotherapy, with precise dose delivery verified through quality assurance measures. Acute toxicity assessment at treatment completion revealed manageable cystitis, with one case persisting at the three-month follow-up. Gastrointestinal toxicity was minimal. For pancreatic tumors, daily adaptation of organ-at-risk (OAR) and gross tumor volume (GTV) was practiced, with median doses to OAR within acceptable limits. Three patients experienced gastrointestinal toxicity, mainly nausea. Overall, the study demonstrates the feasibility and safety of extreme hypo-fractionated radiotherapy on a 0.35T MR-LINAC, especially for challenging anatomical sites like prostate and pancreatic tumors. These findings support the feasibility of MR-LINAC-based radiotherapy in delivering precise treatments with minimal toxicity, highlighting its potential for optimizing cancer treatment strategies.

## 1. Introduction

The recent clinical implementation of linear accelerators integrated with magnetic resonance imaging on board (MR-LINAC) and the consolidation of stereotactic body radiotherapy (SBRT) opens numerous opportunities for improving cancer care. SBRT has enabled the minimization of healthy tissue involvement by treatment margins reduction and an increase in potentially ablative doses to the tumor; due to the recent widely recognized results, this advanced technique was positioned as a frontline strategy in several oncological scenarios, from local to metastatic disease [1]. MR-LINACs allow for daily treatment adaptation and real-time MR-imaging during radiotherapy; both elements come together to deliver safe treatments with improved upfront oncological outcomes and low toxicity rates [2]. This real-time adaptation capability of MR-LINACs allows for the daily re-optimization of the treatment plan, considering variations presented by the patient from planning to each treatment session, demonstrating better coverage of the volumes to be treated, enhanced protection of organs at risk (OARs), and facilitating dose escalation. One of the benefits of using these treatment methods is that they can target tumors located in challenging anatomical areas that are typically difficult to treat with traditional radiotherapy. This is due to the limited tolerance of organs at risk (OARs) to ionizing radiation and the significant mobility of structures such as ultra-central lung tumors or those in the upper abdomen. Further clinical studies are waiting for other locations where dose escalation has shown promising clinical advantages [3,4]. Currently, prostate tumors [5] and pancreatic tumors [6] emerge as locations with the most significant benefit and experience from guided adapted approaches, though the list continues to grow. Reports are promising, with minimal acute toxicity rates that do not impact the quality of life and confirm reasonable local control [4]. In this study, we prospectively evaluate the feasibility and acute toxicity of patients undergoing hypo-fractionated radiotherapy on a 0.35T MR-LINAC. 

## 2. Materials and Methods

### 2.1. Study Design and Patient Selection

The clinical workflow discussed in this article resulted from a collaborative effort by the multidisciplinary team of the Viewray MRIdian^®^ Radiotherapy Unit (ViewRay Inc., Oakwood, GA, USA) in Vithas La Milagrosa Hospital, Madrid. It was developed with the support of the ViewRay Group’s training program and adapted to comply with government regulations and the unit’s daily workflow. The Ethics Committee approved an observational, descriptive, longitudinal, and prospective basket-type study to evaluate acute and late toxicity in patients undergoing hypo-fractionated Magnetic Resonance Guided Radiotherapy (MRgRT) (Registry #5289). Treatment localization toxicity is recorded by Common Terminology Criteria for Adverse Events (CTCAE 5.0) at the end of treatment, 15 days, 3 months, 6 months, 9 months, 1 year, and 2 years. Recruitment for the study began on 1 August 2023, with an estimated ending date of 1 August 2026, after obtaining informed consent from all participants. 

### 2.2. Pretreatment Workflow 

Patients who met the eligibility criteria for treatment on the 0.35T Viewray MRIdian^®^ were considered for inclusion in this research study. The inclusion criteria required a histologically confirmed diagnosis of solid tumors, a Karnofsky Performance Status score greater than 80%, an age over 18 years, and an indication for receiving hypo fractionated or SBRT radiotherapy treatment. Patients who lacked histological confirmation of tumor pathology, were pregnant or breastfeeding women, and those with a history of diseases that cause DNA repair failures, such as Xeroderma pigmentosum, ataxia-telangiectasia, and Fanconi anemia, were excluded. Given our center’s reach across Spain, the potential candidates underwent a thorough review by a multidisciplinary committee before being accepted into the facility. Additionally, patients were required to receive outpatient care and undergo a safety screening using the Magnetic Resonance Imaging Safety Evaluation Form before treatment simulation [7].

#### Simulation Protocol

On the same day, all patients were simulated using computed tomography (CT) and magnetic resonance image (MRI) simulation. Body coils are essential for image acquisition in MRI simulations. They must be close to the treatment target for a high-quality image, apparent distortions, and minimal respiratory movements. Custom-designed prismatic glasses allowed patients to follow precise instructions on the screen for deep inspiration breath hold (DIBH), especially for upper abdominal targets.

MRI simulation was acquired directly from 0.35T MR-LINAC. Patients were in the decubite supine position with arms along the body. Leg supports, cushions, and blankets were used to provide comfort. First, a low-resolution scan was obtained to correct positioning and localization of treatment (pelvis, upper abdomen, thorax). Then, True Fast Imaging with steady-state-free precession (TRUFI) was acquired. The TRUFI sequence was the base sequence for planning and movement gating. This pulse sequence was a balanced steady-state free precession (Bssfp), yielding a T2/T1-weighted contrast. In this step, the physician and physicist determined the localization of treatment, the isocenter, and the target (e.g., the prostate in prostate cancer patients). The target served two purposes: to determine the isocenter and to determine the organ selected for gating. 

MR cine image was the last step of the simulation. The Viewray MRIdian^®^ could perform axial, coronal, and sagittal cine images to follow the target volume previously designed by the physician. We commonly used sagittal views, but a coronal MR cine was selected for gating at the physician’s discretion in upper abdominal cases. [8] It was important when evaluating gating that the patient followed the instructions on how to perform DIBH. 

For prostate tumors, patients were required to have a full bladder and an empty rectum. A week before treatment, patients received a nurse consultation explaining an astringent diet and bowel regulation with laxatives. To better contour the urethra, a urinary catheter was placed on the day of the simulation. Upper abdominal targets were treated with at least a 4 h fast, and Scopolamine butyl bromide was administered before every session to distinguish bowel movement. In some cases, patients were encouraged to drink water 15 minutes before entering to facilitate duodenum contouring in adaptation. 

The Conotur Protégé AI+^TM^ (MIM Software^®^, Version 1.1.3, Cleveland, OH, USA) contouring system, which radiation technicians validated, was used to contour OAR on the TRUFI sequence. Gross tumor volume (GTV) and clinical target volume (CTV) delineation followed international clinical guidelines for the treatment location. To maintain consistent MRI contouring, we established a peer-review system that included a senior radiation oncologist partner and a radiology physician. Additionally, the MIM Software AI^®^ (Version 1.1.3) enabled physicians to fuse previous CT, MRI, and positron emission tomography (PET-CT) performed by the patient using either a rigid or deformable fusion. This approach guaranteed that our radiation therapy was as accurate and targeted as possible.

### 2.3. Treatment Planning 

After the previously contoured TRUFI sequence, which included OAR and GTV or CTV, was transferred to the Treatment Planning System (TPS), a second validation process was performed to ensure no overlapping clinical structures. This system was fully integrated into the Viewray MRIdian^®^ for delivery treatment. Planning Target Volumes (PTV) were created using Boolean operators following clinical protocol. For example, GTV plus 2 mm generated the PTV in prostate cancer treatment, and GTV plus 3 mm generated the PTV in other localizations. No Planning Organs at Risk (PRV) were created; therefore, to optimize planning treatment, physicists used Boolean operators to design two PTV new structures keeping in mind the dose OAR limiting structures or clinical structures (CS): *PTVhigh* = *PTV* − *(CS* + 2 mm*)**PTVlow* = *intersection between PTV and (CS* + 3 mm*) in pancreatic patients*


This process ensured that the treatment area was precise and targeted, minimizing damage to healthy tissue. The voxel image obtained from the Viewray MRIdian^®^ was restricted, which meant that the PTV margins had to be consistent with the grosser of the TRUFI sequence obtained. This ensured that the margins were appropriate and consistent throughout the treatment.

Viewray MRIdian^®^ has two planning systems based on a Monte Carlo algorithm optimization with different inherited weight optimization systems. The first one is the initial one, the planning, where the clinician and the physicist agree on what is achievable or a robust plan for the patient. This plan had parameters and rules for the structures needed to reach the desirable objectives. The second plan, explained below, recalculated the initial planning with the differences between the initial image and the on-day MRI. 

Fractionation schemes and OAR constraints were designed considering localization, previous radiation, and tumor volume within the study protocol detailed in Appendix A. The selection of the fractionation scheme was left to the physician. 

The planning objective was for 95% of the PTV to receive ≥95% of the prescribed dose and 98% of PTVhigh to receive ≥95% of the prescribed dose. If the mandatory OAR constraints could not be met, PTVlow coverage was reduced until the constraints were met. The objective was to obtain a robust treatment plan with a Step and Shoot intensity-modulated RT (IMRT) through a system of penalty functions that could permit a fast daily adaptation during the online clinical workflow. A Monte Carlo algorithm performed dose calculation.

### 2.4. Online Clinical Workflow 

The 0.35 MRI Linac accelerators workflow was previously described by Klüter et al. [8]. Figure 1 represents our online workflows and the department responsible. On treatment day, the patients passed an MRI safety check before entering the room and were placed in the same position as the simulation. The pilot was obtained first for setup and position confirmation. Then, a new on-day TRUFI sequence was acquired and compared with the simulation TRUFI sequence. Similar to cone-beam computed tomography (CBCT) in conventional LINACs, movements to the couch were sent to the MRI Linac with the correction movements in order to fit on-day anatomy into the simulation anatomy. The most important structure to match was GTV or CTV, depending on the treatment.

After couch movement, the physician performed a second review of OAR and GTV with a 3 cm ring originating from the GTV in the on-day TRUFI. If necessary, the physician recontoured the OAR and GTV to adapt to the changes of the day at his discretion. In cases where constraints were limited by the median dose of volume dose in percentage, the whole OAR had to be modified even if it was outside the 3 cm ring. 

The approved OAR and GTV were run by the second planning system. This calculation was almost the same as that used in the initial plan, with three exceptions. The maximum number of voxels used in the optimization of each structure was 65536. If there were more voxels, the structure was re-resampled. The skin was a particular contour in which one of each of the eight voxels was considered for optimization. In the cost function, only the structures inside the beams were considered, but all structures were reported later in the final calculus. This is the main reason for being consistent with the margins.

Clinicians and physicists reviewed the plan. A manual plan was performed if they did not meet our OAR constraints or PTV coverage by protocol. A gating boundary of 2 mm around a gating region of interest (gROI) following the PTV was commonly used (range: 2–3 mm), with 80% of the gROI (range: 75–90%) required to be within the gating boundary for the beam to engage automatically.

### 2.5. Quality Assurance 

We performed two patient-specific Quality Assurances (Qas) for the simulated and daily adaptation plans. The Viewray MRIdian^®^ included the first one with a Monte Carlo calculation engine of dose and Monitor Units (MUs). The second was ArcCHECK^®^-MR (detector array), where we calculated the dose distribution and compared it with the administered plan [9].

### 2.6. Statistics

Descriptive statistics summarize the patient characteristics, treatment time, patient planning details, and acute toxicity. Absolute and relative frequencies to express qualitative variables and the confidence interval of the percentage are also included to depict the dispersion of the results. Concerning quantitative variables, their parametric behavior was assessed, indicating the mean and standard deviation if they followed a normal distribution and the median and interquartile range otherwise.

## 3. Results

Thirty-seven patients were selected for inclusion in this study between August 2023 and December 2023. Patients and tumor characteristics are summarized in Table 1.

A total of 204 fractions were delivered. The median time for the simulation and start of treatment was 6 days. At first, all contours were adapted to Shape into daily TRUFI by deformable deformation based on artificial intelligence. After ten patients, it was decided to use a rigid fusion for GTV, which allowed us to adjust more efficiently.

The timeline for the online adaptive workflow was measured starting with daily TRUFI sequence acquisition and finalizing closing daily treatment. For prostate cases, treatment time ranged from 25 to 45 min, while for upper abdominal lesions, it ranged from 30 to 90 min. All OAR within the 3 cm ring was recontoured in daily adaptations in all patients. In upper abdominal cases, the duodenum and stomach were the most frequent OARs for adaptation mainly because of the size increase. In these cases, GTV was adapted if an overlap with the OAR was present but not because of changes in GTV size. The prostate, bladder, and rectum were adapted in all patients on a daily basis. Seminal vesicles were the most affected by bladder filling.

Manual planning was performed in 83.2% of the fractions delivered. Predicted planning was performed in 16.8% of all prostate cases.

To reduce time in the online workflow, a second clinician was encouraged to be present during upper abdominal cases for contouring and DVH validation.

The median age was 71 (45–84) years, with the majority of patients being male at 75.68% (*n* = 28 patients) and women at 24.32% (*n* = 9 patients). Most patients (68.48%) had a Karnofsky Index of 100. Localizations treated were prostate (59.46%), pancreas (32.44%), adrenal metastases (2.70%), liver metastases (2.70%), and lung metastases (2.70%; Table 2; Appendix A).

Twenty-two prostate patients presented with a clinical tumor T1 (13.63%). The other tumors were T2a (36.36%), T2b (18.18%), T2c (27.27%), and T3a (4.56%). Intermediate favorable risk was the most frequent, found in 50% of the patients. The median PSA was 7 ng/mL (range 1.25–14 ng7ml), the median prostate volume was 39 cc (range 17–123 cc), noduled were most frequently located in left lobule (40.91%), the prescription dose was 36.25 Gy in five fractions of the prostate gland in low and intermediate favorable risk and 40 Gy in five fractions of the prostate gland plus 2 cm of seminal vesicles in intermediate unfavorable risk and high risk by every order day. The median PTV dose was 38.95 Gy, and the median PTV was 95% to 93%. At the end of treatment, nine patients presented with grade I cystitis (40.91%), and four presented with grade II cystitis (18.18%), with no patients with grade III or more. Two weeks after radiation, one patient persisted with grade II cystitis in remission after receiving steroids and nonsteroidal anti-inflammatory drugs. No acute gastrointestinal (GI) toxicity related to treatment was recorded. 

A single patient diagnosed with adenocarcinoma prostate with favorable intermediate-risk was not included in the ultra-hypo-fractionated group due to having a prostate volume greater than 100 cc, which was determined with the MR T2 image. Instead, this patient was treated with a moderate hypo-fractionation of 60 Gy in 20 fractions. The patient finished treatment with grade II cystitis that persisted for 2 weeks after treatment and no GI toxicity (Table 3). 

Twelve patients with locally advanced nonresectable pancreatic cancer were treated; nine patients were cT4(75%), and three were cT3(25%). The most frequent was no nodal involvement in 75% of the patients. All patients were treated with DIBH. The target volume was the gross tumor volume contour, a 3 mm margin, and no elective clinical nodes. Eight patients (66.67%) received systemic treatment with FOLFIRINOX with at least five to ten cycles (83.33%). An ablative dose was delivered with a median prescription of 42 Gy (range 30–50 Gy). The dose coverage was a median of 92% of the PTV of the 95% of the dose prescribed (median Dmean = 40.67 Gy; median Dmax = 48.21 Gy). Daily adaptation of the OAR and GTV was performed in all patients. There were median doses to OAR constraints (Duodenum: median V36 Gy = 0.03 cc; V33 Gy = 0.14 cc; V25y = 2.76 cc). Three patients had treatment for gastrointestinal toxicity grade I (*n* = 2) and grade II (*n* = 1) nausea. A 63-year-old woman presented stage IV adenocarcinoma of the pancreas, which was treated in another center with 60 Gy in 15 fractions in 2020; after a stable response, the patient showed an in-field recurrence. She was treated with 30 Gy in five fractions [PTV (V95% = 95.42%; Dmedian = 31.39 Gy; Dmax = 33.61 Gy); PTV low (Dmin = 22.52; Dmedian = 28.65; Dmax = 32.32)] with OAR contains [Stomach (V25 Gy = 0 cc; V20 Gy = 0.04 cc; V15 Gy = 1.83 cc) Duodenum (V25 Gy = 0.07 cc; V20 Gy = 0. 22 cc; V15 Gy = 0.60 cc) Bowel (V25 Gy = 0.00 cc; V20 Gy = 0 cc; V15 Gy = 0 cc)] adapted to the previous treatment. At a three-month follow-up, the patient presented with symptoms of abdominal pain and was diagnosed with a duodenal ulcer grade III using endoscopy (Table 4).

A female patient was previously treated in 2021 with an oligorrecurence by a colon adenocarcinoma with five liver metastases with our CyberKnife VSI^®^. Doses were prescribed at 76% isodose of 50 Gy in five fractions after a complete response. In October 2023, the patient presented with a single oligoprogression in segment I. The patient refused new fiducial markers, and an MgMRT was prescribed for the PTV 40 Gy in five fractions with no toxicity and liver function alterations at 3 months.

A 79-year-old patient with prostate adenocarcinoma treated in 2007 with brachytherapy presented nodal recurrence with an intermediate unfavorable risk posteromedial zone. A 30 Gy in five fractions to the whole prostate gland was administered. The presence of the urethra stenosis meant that the patient underwent a cystostomy before radiation and removal 2 weeks after, and at 3 months, no acute GU and GI toxicity presented.

One patient was treated for peripheric lung metastases from lung carcinoma at one fraction of 28 Gy in DIBH. The PTV coverage was 96.55%, D media was 31.01 Gy, and Dmax was 35.72 Gy. Before treatment, the patient presented with moderate dyspnea, which was not modified after 3 months of treatment.

Regarding quality assurance results, patients showed a gamma (2%, 2 mm) >99% in the secondary calculation of dose and MU. Second, during the QA verification, patients achieved a gamma (2%, 2 mm) above 98% with a threshold of 15% for prostate cases and at least a gamma (3%, 2 mm) with a threshold of 15% >95% in treatments with more stringent dosimetry (e.g., pancreas, lung).

## 4. Discussion

This article presents a clear and feasible protocol for an MRI-guided radiation workflow that considers the unique Spanish legislation on radiation delivery, management, human resources, and scientific literature available on this topic. In July 2023, a significant milestone was reached for our team in Spain with the MRIdian^®^ MR-Linac successfully treating a prostate cancer patient via five-fraction SBRT MRI-guided Radiotherapy. The treatment was adapted daily in each fraction, and there were no signs of toxicity upon evaluation 3 months later.

Introducing this groundbreaking technology presented numerous challenges and opportunities in the market, operational, and clinical spheres. As Hehakaya et al. (2022) [10] note, emergent medical technologies, realignments in workflows, radiotherapy reimbursement issues, socioeconomic considerations, and patient apprehensions influence MR-Linac adoption practices across diverse healthcare systems and nations. In Spain, the National Health System (NHS) primarily provides radiation oncology services, setting the standard for radiation practices across the nation. Implementing recent technologies in the private sector, as in our case, can be challenging due to the lack of established practices within the NHS.

For this reason, our market strategy assumed that we would be more closely aligned with the clinical patterns observed in the United States, in the majority of private-sector providers, than in Europe, as published by Chuong et al. [11]. They outlined the clinical adoption patterns of MR-Linac in the United States between 2014 and 2020, with pancreatic malignancies (20.7%), liver tumors (16.5%), and prostate cancer (12.5%) being the most frequently treated pathologies. Interestingly, they observed a rising trend in the indication rates for pancreatic cancers and prostate malignancies in 2020, with a propensity for over a 15% increase. Our results with 59.46% prostate cases and 32.44% pancreas cases align more with the clinical patterns observed in Europe reported by Sloetman et al. [12], with an annual growth trajectory in pancreatic tumors (157.1%), liver malignancies (134.2%), and prostate cancer (120.9%) being the most frequently treated pathology (23.5%), followed by pancreatic malignancies (11.2%). One reason for this may be the comprehensive review performed by our clinicians on the clinical indications in MRI-guided radiotherapy with the assurance that patients would benefit from MRI gating with intra-fraction motion management, online and offline adaptation, and improved soft tissue visualization [13]. This allows us to reach out directly to our referrals and creates opportunities for collaborations with the NHS.

The creation of evidence-based protocols for prostate MgmRT and pancreas MgmRT was the first encounter for all departments with this technology. We based the role of MRgRT in prostate cancer on several publications, such as Kishan et al. [5] in the phase III MIRAGE trial. This trial compared prostate SBRT in MRI guidance to CT guidance. The results showed that there were 0.0% acute grade II or more GI toxic effects with MRI guidance compared to 10.5% with CT guidance, even though all patients received radiation to the prostate and 1 cm of seminal vessels at 40 Gy in five fractions with no discrimination between risk baseline population. Additionally, the GU toxicity was lower by 24.4% with MRI guidance compared to 43.4% with CT guidance [5]. Teunissen et al. [14] presented the 12-month follow-up results of the MOMENTUM study, highlighting an increase in GU and GI grade II at the 3-month evaluation with 23.8% and 5%, respectively, among 82% of intermediate-risk patients treated with 36.25 Gy. Alongi et al. [15] investigated quality of life and patient-reported outcomes measures (PROMs) in patients treated with SBRT 35 Gy in five fractions. There were no differences in the EORTC Quality of Life Questionnaire-Core 30 (EORTC QLQ-30) after treatment and no grade III toxicity.

In the case of radiation for the pancreas, the role of conventional radiation therapy (CRT), in combination with chemotherapy, fails to improve overall survival (OS) in the two major clinical trials, LAP07 [16] and CONKO-007 [17]. Tchelebi et al. [18], in a metanalysis, compared the OS between CRT and SBRT in locally advanced pancreatic cancer (LAPC), favoring SBRT with an increase in the OS in 27% vs. 14% at 2 years. To our knowledge, no phase III trial compares CT-based SBRT in MRI guidance and CT guidance.

Hassanzadeh et al. [19] reported the first experience of stereotactic MRI-guided radiotherapy (SMART) in inoperable pancreatic cancer treated with 50 Gy in five fractions daily. They observed late gastrointestinal grade II toxicity of 4.6% and no acute toxicity, with an overall survival rate at one year of 68.2%. Ruda et al. [20] compared MRgRT with conventional fractionation, hypo-fractionation, and SMART, noting a statistically significant improvement in overall survival at 2 years by 49% compared to 30% for ablative dose with intriguing toxicity grade III in 3 patients with conventional radiotherapy and non-SMART treatments. This favors the hypothesis that high-dose radiotherapy improves overall survival and has lower toxicity. Since then, two phase II publications of SMART have been published with grade II acute toxicity (gastrointestinal ulcers) ranging between 2.9% and 8.8% [21]. 

The logistical procedure implementation of MRgRT truly represented a game-changer in our daily operational workflow. On one side, we are faced with an Official State Law since 2006, dictating that radiation oncologists are responsible for contouring OARs, GTVs, CTVs, PTVs, prescriptions, and treatment administration, while on the other side, we aim to be efficient with our human resources [22]. A solution was creating a new position for contouring and daily adaptation for clinicians and the physicist department, as Lamb J et al. [23]. described, with the concept of a “doctor of physicist on the day.” However, this presented a significant disadvantage for us as it did not ensure a fast learning curve for our staff to achieve the optimal goal of reducing treatment and simulation times. To address this, we leveraged the advantage of the Viewray MRIdian^®^, which can facilitate simultaneous work among different stations during daily adaptation. 

RTT simulation protocols had to be created with a unique perspective in mind. Typically, in CT guidance radiation, the CT simulation is not conducted on the same machine that is used for delivery. Therefore, time management for the Linac had not to be divided between simulation and daily treatment. Although our simulation median time is half an hour, we allocate the same time slots for simulation as for treatment, considering that SBRT is prescribed on alternate days or daily. From that, we plan simulation slots accordingly.

Another important consideration is that a full bladder is mandatory during treatment for prostate patients in CT guidance radiotherapy. Initially, this led to interruptions during simulation or treatment. To counter this, we established the practice of starting treatment with half a bladder with 250 mL of water [24]. Also, to minimize the risk of acute GU toxicity, it is recommended to clearly identify the urethra, as demonstrated in the PACE B study [25]. This allows for better control of hotspots with a Dmax less than 42 Gy. While Pham et al. [26] have evaluated two MRI sequences–3D HASTE and 3D TSE–in a 0.35 T MR-Linac and concluded that 3D HASTE is the superior sequence, we believe it is more prudent to start our learning curve by beginning with an MRI simulation with bladder catheterization and transitioning to MRI simulation with 3D HASTE.

For physicists, one of the disadvantages of MRgRT is the longer treatment time for Beam ON, which is due to the dose rate and the Step & Shoot technique. For this reason, Grimbergen et al. suggested that reducing intrafraction motion could be achieved with a corset during MRgRT in localization, where respiratory cycles play a key role in tumor margins and dose prescriptions. They concluded that there was no correlation between respiratory movement and dosimetry impact. Still, they advised that this represents a limited view of the respiratory cycle and does not consider larger respiratory cycles [27]. Therefore, we still perform DIBH in the thorax and upper abdominal treatments, and physicists design a robust modulation plan with a time limit of less than 15 min on Beam ON.

Clinicians and RTTs, with a second verification by clinicians, readapt contours and target structures at two different stations while physicists delineate air and water structures at a third available station. These stations could be arranged side by side, or, as in our clinic, two are in the Linac’s room and another in a consultation room. Simultaneous work allows us to maintain treatment times similar to those described in the literature, with a median of 39 (22–59) minutes for pancreatic cases, even though our radiation oncologists have to verify the contours of RTTs, and daily re-optimization was performed in all patients [28]. Additionally, we developed a rotating schedule among our clinicians that allows them to participate in the MR-Linac workflow, with adaptations comparable to the first clinical consultation, thus increasing operational efficiency.

When creating a rotation schedule, we had to consider the daily inter-observer variability in GTV and OAR, which could potentially impact planning coverage, toxicity, and, ultimately, local control results. Smith et al. [29] conducted an offline study in 2023 to address this issue. They compared the contouring of five radiographers and five radiation oncologists with prostate and seminal vessel contouring on ten MRIs acquired by an MR-Linac. The study found significant differences in the apex and base contours of the prostate. However, there was no statistical difference in coverage during planning. The study concluded that the base image is better suited for contouring the prostate, compensating for inter-observer variability in contouring, while bladder and rectum identification were unaffected. We attempted to minimize this variability with a peer-review system: before adaptation (radiation clinician with a radiologist clinician), during adaptation (clinician responsible for treatment and a second clinician), and after treatment with daily communications between our group, with the day’s specifics. We are waiting for our internal results on this subject.

Over 5 months had passed, and 204 fractions were delivered in a 0.35T MR-LINAC. Our population consisted of 37 patients between hypo-fractionated (2.7%) or SBRT radiotherapy (97.3%) with heterogeneous. The advantage of daily adaptation and motion management presented us with several unique cases.

Three patients in our cohort presented with high-risk prostate cancer (HRPCA) defined by NCCN Clinical Practice Guidelines in Oncology [30]: Prostate Cancer 2023 (cT3a and Gleason Score 8 and/or PSA ≤ 20 ng/mL). These three patients were treated with 40 Gy in five fractions to prostate and seminal vesicles and no pelvic irradiation. One patient presented grade II GI, and their treatment was prolonged up to the 3-month evaluation in remission with steroids. To our knowledge, using SBRT in high-risk patients could be a possible treatment in selected patients. In CT-guided SBRT, two prospective studies explore this subject. The HYPO-RT-PC-TRIAL, a phase III trial, included 11% of the patients (HRPCA) in the arm of 42.7 Gy in seven fractions, with a grade II–IV in 28% of the patients’ GU. However, it showed a similar late toxicity and biochemical control at 5 years of 84% [31]. The FASTR-2 explored lowering the doses to 35–40 Gy in five weekly fractions without pelvic radiation. It showed acute grade II GU at 14.8%, GI at 3.7%, and only grade II toxicity in GU at 21.7% [32]. MR-guided SBRT in HRPCA is presented in a single-arm phase II study conducted by Bruynzeel et al. [33], in which 59.4% of patients were treated with 36.25 Gy in five fractions to the prostate and base of the seminal glands. The study hypothesized a lower acute GU toxicity of 40% compared to the literature of hypo-fractionated radiotherapy in 61%, with better results possibly due to urethra sparing by daily adaptation in 23.8% and acute grade II GI toxicity in 5% with pending validation on the ongoing phase II SMILE trial [34]. Another consideration in these patients is the biochemical control (BCR) in not adding pelvic node radiation to SBRT in HRPCA. This is controversial because of the lack of consistency in the studies and subpopulation analysis. In favor of pelvic node radiation, the SATURN trial [35] reported a BCR of 100% at 2 years, while the HYPO-RT-PC trial, pHART 8 trial [36], and FASTR-2 trial demonstrated BCR rates of 84% and 85.4% at 5 years, with 100% observed within the first year of follow-up with nonpelvic nodes radiation. We decided not to irradiate pelvis nodes because none of the patients presented with very high-risk prostate cancer.

A patient with good performance status met the criteria for local prostate recurrence after a biopsy-proven nodal recurrence 16 years following a low dose rate of brachytherapy of the entire gland. In 2018, the Australian and New Zealand Radiation Oncology Genito-Urinary group suggested that salvage local radiotherapy should be considered for patients with a life expectancy exceeding 10 years [37]. Due to urethral stenosis complications from previous treatment and contraindications for other local treatments due to comorbidities, SBRT emerged as a viable option with the lowest rate of >grade II GU toxicity, at 4.2% (95% CI: 0.8–9.1%), and a GI toxicity rate of 1.9% (95% CI: 0.6–3.7%) among other local techniques [38]. It could be argued that because of the presence of urethral stenosis in the patient, a focal reirradiation to the prostate should be performed. Still, as signaled in the MASTER STUDY, all available evidence of SBRT is reirradiation to the whole gland. With this precedent and the integration of high-resolution MRI simulation and gating, it is conceivable that MRgRT will further reduce these percentages and could possibly open the possibility of a clinical trial comparing focal radiation to whole prostate gland radiation in this setting [39]. We opted to implement the technique with a prescription of 30 Gy to the entire gland using a urethra-sparing approach, resulting in no acute toxicity. A similar approach to prescription using MRgRT was described by the Montpellier Institute in 2022, yielding comparable results and a biochemical control rate at one year of 65% [40].

Lominska et al. [41], Wild A.T. et al. [42], and Koong et al. [43] discuss the possibility of reirradiation of the pancreas with SBRT after conventional treatment with CyberKnife, with local failure rates ranging between 12% and 20% at 12 months. Among the 61 treated patients, 10 presented > grade III toxicity, less than 10% when treated in multifraction (5 fractions), with doses ranging from 25 to 33 Gy in 5 fractions. Bryan et al. [44], in 2020, described the first series of cases of upper abdominal reirradiation with MRIdian^®^ MR-Linac. Similar to us, the most frequent localization treated in this study was lymph nodes and recurrent pancreatic cancer; intriguingly, the authors prescribed doses with an EQD2_10_ of 40–50 Gy without considering that OAR contouring diminishes margins in PTV and gating, with no evidence of grade III toxicity. In our patient, because previous radiation treatment was 60 Gy in 15 fractions to preserve constraints and limitations for reirradiation, we opted for conventional doses of 30 Gy in 5 fractions.

Our study has limitations, such as a small sample size, a short follow-up period, and varied treatment plans. Considering our limitations, we can see that we are already below the toxicity numbers described in the literature. In prostate MgmRT, GU and GI grade II toxicity were 18.18% and 0%, respectively, with no increase at the 3-month follow-up. In pancreas treatments, acute grade II toxicity was only presented in reirradiation scenarios and not in primary tumors.

However, our research provides the first insight into this technology in Spain. To improve our findings, we recommend longer follow-up periods to better understand this treatment’s late toxicity and local control.

## 5. Conclusions

MRgRT represents a novel approach in Spain, with the ability to adapt OARs, leading to improved clinical outcomes and reduced toxicity. This article aimed to analyze this technique’s feasibility and clinical toxicity. We encountered numerous challenges and new opportunities in integrating this technology. The implementation has been successful, with acute toxicity rates consistent with the literature. In summary, the results suggest that the introduction of MRI-guided radiotherapy is feasible and safe.

## Figures and Tables

**Figure 1 cancers-16-01685-f001:**
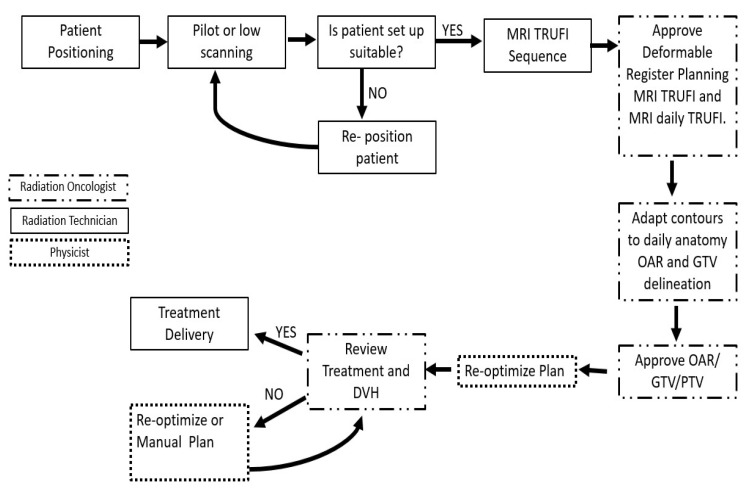
Clinical online workflow of daily adaptation in MR-Linac. Abbreviations: DVH = dose-histogram-volume; GTV = gross tumor volume; MRI = magnetic resonance; OAR = organs at risk; PTV = planning target volume.

**Table 1 cancers-16-01685-t001:** Patient characteristics. Cht = chemotherapy; GTV = gross tumor volume.

Characteristic	Value
**Median Age** **(range)**	71 (46–84)
**Gender (%)**	
Male	28 (75.68%)
Female	9 (24,32%)
**Karnofsky (inclusion ≥ 80%) (%)**	
80	1 (2.70%)
90	6 (16.22%)
100	30 (68.46%)
**Localization treatment (%)**	
Prostate	22 (59.46%)
Liver	1 (2.70%)
Pancreas	12 (32.44%)
Adrenal	1 (2.70%)
Lung	1 (2.70%)
**Reirradiation (%)**	
Yes	3 (8.10%)
No	0 (91.9%)
**Prostate: cT-stage (%)**	
T1	3 (13.63%)
T2a	8 (36.36%)
T2b	4 (18.18%)
T2c	6 (27.27%)
T3a	1 (4.56%)
**Risk-stage according to NCCN guidelines (%)**	
Low Risk	3 (13.64%)
Intermediate favorable risk	11 (50%)
Intermediate unfavorable risk	5 (22.73%)
High risk	3 (13.64%)
**GTV (%)**	
Prostate	14 (63.64%)
Prostate and seminal vesicles	8 (36.36%)
**Pancreas: cT-stage (%)**	
T3	3 (25%)
T4	9 (75%)
**cN-stage (%)**	
N0	8 (66.67%)
N1	4 (33.33%)
**Systemic treatment**	
FOLFIRINOX	9 (75%)
Other	3 (25%)
**Chemotherapy cycles n (%)**	
<5	1 (8.33%)
5–10	10 (83.33%)
>10	1 (8.33%)

**Table 2 cancers-16-01685-t002:** Localization and fractionation scheme. Fx = fractions; D = daily; AD = alternate days.

Localization	Patients	Dose	Target Volume
Prostate	14	36.50 Gy in 5 fx. (AD)	Prostate
	8++	40 Gy in 5 fx. (AD)	Prostate and seminal vesicles
Pancreas	3	50 Gy in 5 fx. (D)	Pancreatic tumor
	1	45 Gy in 5 fx. (D)	
	7	40 Gy in 5 fx. (D)	
	1	30 Gy in 5 fx. (D)	
Lung	1	28 Gy in 1 fx. (D)	Lung nodule
Liver	1	50 Gy in 5 fx. (D)	Liver nodule
Adrenal gland	1	36 Gy in 3 fx. (D)	Adrenal gland

**Table 3 cancers-16-01685-t003:** Toxicity assessment MgmRT prostate. CTCAE classification.

Adverse Event	End of Treatment			Three Months		
	Grade I	Grade II	Grade III	Grade I	Grade II	Grade III
Genito-urinary						
Cystitis	9 (40.91%)	4 (18.18%)	0	0	1 (4.54%)	0
Hematuria	0	0	0	0	0	0
Urinary incontinence	0	0	0	0	0	0
Urinary retention	0	0	0	0	0	0
Gastrointestinal						
Diarrhea	0	0	0	0	0	0
Colitis	0	0	0	0	0	0
Rectal pain	0	0	0	0	0	0

**Table 4 cancers-16-01685-t004:** Toxicity assessment MgmRT Pancreas. CTCAE classification.

Adverse Event	End of Treatment			Three Months		
	Grade I	Grade II	Grade III	Grade I	Grade II	Grade III
Nausea	2 (8.5%)	1 (4.25%)	0	0	0	0
Vomiting	0	0	0	0	0	0
dyspepsia	0	0	0	0	0	0
Jaundice	0	1 (4.25%)	0	0	0	0
Diarrhea	0	0	0	0	0	0
Colitis	0	0	0	0	0	0
Duodenal ulcer	0	0	0	0	0	1 (4.25%)

## Data Availability

The data presented in this study are available in this article and Appendix A.

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
