# Peer review of "Feasibility and Acute Toxicity of Hypo-Fractionated Radiotherapy on 0.35T MR-LINAC: The First Prospective Study in Spain"

_cancers, 2024, doi:10.3390/cancers16091685_

Round 1

Reviewer 1 Report

Comments and Suggestions for Authors

Objective and Relevance: The study aims to evaluate the feasibility and acute toxicity of hypo-fractionated radiotherapy using a 0.35T MR-LINAC, focusing on challenging anatomical sites like prostate and pancreatic tumors. The subject is relevant and timely given the advancements in radiotherapy techniques and the importance of minimizing toxicity in cancer treatment.

Study Design: The observational, descriptive, longitudinal, and prospective design is suitable for the research objectives. The inclusion and exclusion criteria are well defined.

Statistical concerns:

1. Table 1 should be re-format before publishing;

2. The paper mentions using the Kolmogorov-Smirnov test for assessing normality but does not provide details on how the findings from this test influenced the choice of statistical tests for further analyses. It's crucial to specify if non-parametric tests were used for data not following a normal distribution;

3. The paper could benefit from a more detailed description of the treatment planning and adaptation processes. This would help in understanding how daily adaptations were implemented and their impact on treatment outcomes.

Author Response

Dear Reviewer: 

Thank you for your report. Here are the comments and rearrangements 

1. Table 1 should be re-format before publishing: Response; Table 1 is re format for consideration

2. The paper mentions using the Kolmogorov-Smirnov test for assessing normality but does not provide details on how the findings from this test influenced the choice of statistical tests for further analyses. It's crucial to specify if non-parametric tests were used for data not following a normal distribution; Response: as is a descriptive study we use percetanges and median. The test is going to be used for futher publication

3. The paper could benefit from a more detailed description of the treatment planning and adaptation processes. This would help in understanding how daily adaptations were implemented and their impact on treatment outcomes. Response: We rewrite the section of better understanding 

Your Sincerly, 

Daniela 

Reviewer 2 Report

Comments and Suggestions for Authors

Thank you for giving me opportunity to review the manuscript.

This paper is an interesting study on the introduction of 0.35 MR Linac in Spain. Although it showed good results in terms of acute toxicity, results on treatment outcomes could not be shown due to the short study period. To explain the feasibility of MRI-guided RT, a more detailed explanation of radiation therapy is needed.

In Materials and methods

1) The study design is described as hypo-fractionated MRgRT. What are the fraction sizes and number of fractions which allowed in the original study design?

2) The PTV margin for prostate cancer is mentioned as GTV+2mm. Was the PTV margin set the same for the remaining organs - pancreas, liver, adrenal, and lung?

3) What were the constraints for each critical organ in the treatment plan?

In Result

1) What are is criteria for selecting SBRT or moderate hypofractionation according to prostate volume?

2) It is necessary to add a Table detailing radiation therapy dose, fraction size, target volume, and OAR constraints.

3) Was on-line adaptation performed for each fraction in all patients? and how many factions actually had adaptation performed after daily registration?

4) Additionally, a detailed explanation is needed as to what the specific reasons for adaptation are (change in volume or position of GTV, change in critical organ, etc.).

5) According to "Toxicity" paragraph, Grade 3 GI toxicity associated with RT were observed in one patient. Of this patient, the description of the dose-volume parameters of involved organs is of value.

Author Response

Dear Reviewer, 

Thank you for  you thoughtful remarks. I hope we accomplish to satisfy your  amendments.  

In Materials and methods

1) The study design is described as hypo-fractionated MRgRT. What are the fraction sizes and number of fractions which allowed in the original study design? We added a table in a supplementary appendix of fractions scheme available in the study. 

2) The PTV margin for prostate cancer is mentioned as GTV+2mm. Was the PTV margin set the same for the remaining organs - pancreas, liver, adrenal, and lung? GTV`+ 3mm is used for other locatization. We not create a  PRV  because for optimazation we have to PTV volumes( PTV low and PTV High). We introduce this concept now in the articule for your consideration 

3) What were the constraints for each critical organ in the treatment plan?

We made a table detailing the constrains used in the Supplementary Appendix

In Result

1) What are is criteria for selecting SBRT or moderate hypofractionation according to prostate volume? Yes, by protocol we don`t treat volumenes higher tha 100 cc. 

2) It is necessary to add a Table detailing radiation therapy dose, fraction size, target volume, and OAR constraints.

We added a table for consideration

3) Was on-line adaptation performed for each fraction in all patients? and how many factions actually had adaptation performed after daily registration?

We did adaptation in for all patients because we had the intention of doing a learing curve for MRI contouring. We are now trying to address this  subject in new studies. 

4) Additionally, a detailed explanation is needed as to what the specific reasons for adaptation are (change in volume or position of GTV, change in critical organ, etc.).

We added this question in our results for your consideration

5) According to "Toxicity" paragraph, Grade 3 GI toxicity associated with RT were observed in one patient. Of this patient, the description of the dose-volume parameters of involved organs is of value.  We added  these for your consideration.